# The Many Faces of CD4^+^ T Cells: Immunological and Structural Characteristics

**DOI:** 10.3390/ijms22010073

**Published:** 2020-12-23

**Authors:** Demetra S. M. Chatzileontiadou, Hannah Sloane, Andrea T. Nguyen, Stephanie Gras, Emma J. Grant

**Affiliations:** 1Department of Biochemistry and Molecular Biology, Biomedicine Discovery Institute, Monash University, Clayton, VIC 3800, Australia; dimitra.chatzileontiadou@monash.edu (D.S.M.C.); Hannah.Sloane1@monash.edu (H.S.); andrea.nguyen@monash.edu (A.T.N.); stephanie.gras@monash.edu (S.G.); 2Australian Research Council Centre of Excellence for Advanced Molecular Imaging, Monash University, Clayton, VIC 3800, Australia

**Keywords:** CD4 T cell, Th1, Th2, Th17, Th22, Treg, Tfh, human leukocyte antigen (HLA), structural biology

## Abstract

As a major arm of the cellular immune response, CD4^+^ T cells are important in the control and clearance of infections. Primarily described as helpers, CD4^+^ T cells play an integral role in the development and activation of B cells and CD8^+^ T cells. CD4^+^ T cells are incredibly heterogeneous, and can be divided into six main lineages based on distinct profiles, namely T helper 1, 2, 17 and 22 (Th1, Th2, Th17, Th22), regulatory T cells (Treg) and T follicular helper cells (Tfh). Recent advances in structural biology have allowed for a detailed characterisation of the molecular mechanisms that drive CD4^+^ T cell recognition. In this review, we discuss the defining features of the main human CD4^+^ T cell lineages and their role in immunity, as well as their structural characteristics underlying their detection of pathogens.

## 1. Introduction

CD4^+^ T cells are key players in the adaptive immune response. Following selection and maturation in the thymus, naïve CD4^+^ T cells migrate to the periphery where they survey for antigens displayed by human leukocyte antigen class II (HLA-II) molecules present on the surface of professional antigen presenting cells (APCs) [1], such as Dendritic cells (DCs) [2], B cells [3], macrophages [4], CD4^+^ T cells and airway and intestinal epithelial cells [5]. HLA-II molecules are composed of an alpha (α) and beta (β) chain (Figure 1), anchored into the cell membrane, that together form the peptide (p) binding groove [1] (Figure 1A,B). HLA-II presents peptides that are longer (>11 residues) than HLA class I molecules (8–10 residues) due to the open-ended conformation of their peptide binding groove (Figure 1A). Circulating CD4^+^ T cells recognize peptide-HLA-II (pHLA-II) complexes through their T cell receptors (TCRs), also comprising of an α and β chain (Figure 1C) [6]. If the peptide presented by the HLA-II molecules is recognized as foreign, then CD4^+^ T cells become activated.

CD4^+^ T cells are predominantly known as helper cells. They provide help by producing cytokines, which can recruit other cells of the immune system to the site of infection to help tackle invading pathogens. CD4^+^ T cells license dendritic cells to assist in the activation of naïve CD8^+^ T cells, which are the killers of the immune system [7]. Furthermore, CD4^+^ T cells help B cells, which predominately make antibodies [8]. The help provided by CD4^+^ T cells promotes somatic hypermutation and enables antibody class switching, allowing for the generation of high affinity antibodies of the most effective isotype for the given infection. Interestingly, recent studies have shown that CD4^+^ T cells can also display cytotoxic capabilities [9,10,11,12,13], suggesting that they can play a more direct role in pathogen clearance.

CD4^+^ T cells are extremely heterogeneous. The presence or absence of costimulatory factors, and the cytokine environment during CD4^+^ T cell activation, drives the expression of lineage specific transcription factors. This facilitates the differentiation of CD4^+^ T cells into a range of subsets [14] characterised by unique cytokine profiles, specific transcription factor expression, and a relatively distinct phenotypic profile (Figure 2). Although initially characterised in mice, several CD4^+^ T cell subsets have been identified and well-characterised in humans over the last 30 years; T helper 1 (Th1), T helper 2 (Th2), T helper 17 (Th17), T helper 22 (Th22), regulatory T cells (Treg), and T follicular helper cells (Tfh) (Figure 3).

In this review, we outline the defining features of each of the six main CD4^+^ T cell lineages. We discuss the CD4^+^ T cell subsets in the context of disease and highlight the role of these CD4^+^ T cell subsets in the resulting immune response. Furthermore, recent advances in structural biology have allowed insight into the interactions between CD4^+^ T cell-derived TCRs and their pHLA-II antigens. To date, the structures of 32 unique TCR-pHLA-II complexes are available (Figure 1D, Table 1), 27 of which have been generated in the last decade, and 13 of those in the last 5 years. Therefore, we can now correlate the CD4+ T cell lineages with the available structures. This combination of molecular and functional characterisation of provides a deep and thorough understanding of this burgeoning family of immune cells.

## 2. T Helper 1 Cells (Th1)

### 2.1. Overview of Th1 Cell Features

T Helper 1 (Th1) cells were first described in 1986 [35], where two distinct populations of murine CD4^+^ T cells were observed with distinct characteristics. The homologous Th1 lineage was subsequently identified in humans in 1990 [36] (Figure 3). Being one of the first CD4^+^ T cell lineages characterised, it remains one of the best understood to date. Th1 cells are predominately identified based on the expression of the transcription factor T-bet and their production of IFNγ, which was first known as macrophage-activating factor [37] (Figure 2), playing a central role in macrophage activation against intracellular bacteria and cancer cells [38,39]. Although many studies have tried to clarify the exact intracellular signalling pathways and epigenetic regulations that commit naïve CD4^+^ T cells to the Th1 lineage [40,41,42], there is still some controversy in the field. None-the-less, the most widely accepted paradigm is that during activation, the presence of IL-12 [43] produced by APCs, drives the down-stream expression of STAT4, which regulates T-bet expression in CD4^+^ T cells [44], subsequently facilitating the production of their signature cytokine, IFNγ [35,45,46,47]. Additionally, Th1 CD4^+^ T cells may also produce IL-2 [35,48], IL-10 [49,50], TNF-α [51] and lymphotoxin [35]. Th1 cells can be found circulating within the periphery, and there is mounting evidence that Th1 CD4^+^ T cells can also express tissue resident phenotypes [52,53] and be found in tissues. IL-2 is known to be important in the generation of resident memory Th1 CD4^+^ T cells in the lungs of mice [52], and protective tissue resident CD69^+^ CD4^+^ Th1 cells can be generated by perinatal vaccination and found resident in the genital tract in humans [53]. Th1 CD4^+^ T cells are multifaceted, playing important roles in antiviral, antibacterial, and anti-cancer immune responses. However, they are also significant contributors to autoimmune diseases.

### 2.2. Antiviral Role of CD4^+^ Th1 Cells

Due to their production of potent antiviral cytokines, Th1 CD4^+^ T cells are important in the protection of humans against a range of viruses including Epstein Barr Virus (EBV) [54,55], Human cytomegalovirus (HCMV) [56], Human Immunodeficiency Virus (HIV) [10], and others [57]. Human Th1 CD4^+^ T cells have been particularly well studied in the context of influenza [48,58]. Studies have shown that IFNγ secreting influenza-specific CD4^+^ T cells can be identified in humans following natural exposure to influenza viruses [59] and the frequency of pre-existing CD4^+^ T cells correlated with decreased viral shedding and decreased disease severity in humans challenged with live influenza viruses [11]. IFNγ secreting CD4^+^ T cells can be generated against a range of influenza proteins, with immunodominant responses towards the Matrix 1 (M1) and Nucleoprotein (NP) of influenza viruses [60,61]. One immunogenic peptide generated from the surface glycoprotein Haemagglutinin (HA), namely HA_306–318_, is able to be presented by multiple HLA-DR alleles [62,63]. Numerous studies have characterised CD4^+^ T cell responses towards this universal HA_306–318_ peptide [15,64,65] including their recognition of the pHLA-II complex [15,16]. Two CD4^+^ TCR structures have been solved in complex with HA_306–318_ presented by HLA-II molecules (Table 1) offering insight on how CD4^+^ TCRs can recognise influenza epitopes. Both structures comprise HLA-DR1 presenting the HA_306–318_ peptide. The first structure was published in 2000 [16] and the second structure more recently in 2020 [15]. The two TCRs (HA1.7 and F11) share the same TRAV, but different TRBV genes (Table 1), and both TCRs bind to HLA-DR1- HA_306–318_ in a canonical docking mode, with a similar number of contacts with the pHLA-II. The CDR2α and CDR2β of both TCRs do not make contact with the HA_306–318_ peptide, but the CDR2β is an important determinant of HLA-DRα specificity contacting the HLA-DR1 molecule. Conversely, the CDR1α of both TCRs, which both are both TRAV8-4*01, contact the peptide, which might suggest a peptide driven TCR bias. Another similarity between these two TCRs are their preferences for acidic residues (Asp/Glu) within the non-germline encoded region of the CDR3α, which interact with the P5-Lys of the peptide. This suggests that the charge specificity has a structurally defined role in recognising the HA_306–318_ peptide.

### 2.3. Antibacterial Role of CD4^+^ Th1 Cells

Th1 CD4^+^ T cells play a role in the immune response against bacteria [66] and this has been largely investigated in various mouse models. However, human Th1 CD4^+^ T cells have been identified in response to pathogens including *Mycobacterium Tuberculosis* [67], *Coxiella burnetti* [68], and *Staphylococcus aureus* [69]. Interestingly, studies of *M. tuberculosis* identified a population of CD4^+^ T cells that could recognise *M. tuberculosis*-derived lipid-based antigens presented by the HLA-like molecule CD1b [70]. These T cells, referred to as germline-encoded mycolyl-reactive (GEM) T cells, are characterised by the expression of the CD4 co-factor, and TRAV1-2/TRAJ9 segments in their TCR. GEM T cells can produce IFNγ upon recognition of the glucose monomycolate (GMM) lipid. The structure of a GEM TCR (GEM42) in complex with CD1b-GMM revealed that the glucose moiety of the lipid was wrapped by the CDR3 loops of the TCR (Figure 4A) leading to high specificity and affinity for the antigen [17].

### 2.4. Role of Th1 CD4^+^ T Cells in Autoimmunity

In the absence of effective regulation, Th1 CD4^+^ T cell responses can target self-peptides, resulting in autoimmune diseases such as Type 1 diabetes (T1D) [71], Rheumatoid Arthritis (RA) [72], and Systemic Lupus Erythematosus (SLE) [72]. Since CD4^+^ T cells predominately recognise peptides presented by HLA-II, and are key players in the development of T1D, it is not surprising that there is a strong association with T1D and particular HLA-II alleles, namely HLA-DR3/4 and HLA-DQ2/8 [73,74]. As such, particular T1D epitopes specific for these HLA-II alleles have been identified and characterised. Th1 T cells in individuals with acute onset T1D are specific for GAD (glutamic acid decarboxylase) and PPI (preproinsulin) epitopes [71]. An insulin derived epitope, namely B_9–23_, has been particularly well characterised in humans. One study has shown that individuals with recent onset T1D produce IFNγ in a dose dependant manner towards the B_9–23_ peptide, and that blocking HLA-II molecules prevented T cell proliferation in a single donor [75]. Others utilised HLA-DQ8-B_9–23_ tetramers to identify CD4^+^ T cells in the peripheral blood of individuals with T1D [76]. To further characterise this epitope, structural studies have been undertaken with one TCR bound to HLA-DQ8 presenting the B_9–23_ peptide [18]. The focus and recognition of CD4^+^ T cells in both human and mouse bear some similarity [18], which is important since understanding the role of CD4^+^ T cells in T1D has been largely focused on the non-obese diabetic (NOD) mouse model. The structure of an insulin peptide in complex with HLA-DQ8 shows similarities with the mouse MHC IAg7 in peptide binding, and similar to MHC IAg7, it has a preference for an acidic residue at position 9 (P9) [18].

Coeliac disease is another autoimmune disease caused by circulating Th1 CD4^+^ T cells which respond to gliadin derived peptides from oral food-derived gluten, and induce inflammation within the mucosa of the gastrointestinal tract [77,78]. Similar to T1D, the expression of particular HLA-II alleles, namely HLA-DQ2 and HLA-DQ8 [79,80], determines susceptibility to the development of coeliac disease. Molecular and structural studies have helped further our understanding of CD4^+^ T cell responses to various gluten antigens. Interestingly, deaminated gluten leads to epitopes with stronger binding to HLA-DQ2/DQ8 as well as to T cells. A total of 13 TCR-gluten-HLA-DQ structures have been solved with different gluten epitopes (Table 1, Figure 4B) [19,20,21,22,23]. The structures show that the TCRs display different docking and recognition mechanisms, even amongst the HLA-DQ2- or HLA-DQ8-restricted TCRs. One interesting and common feature of 10 of these TCRs (D2, JR5.1, S16, T15, SP3.4, S13, L3-12, T316, Bel502, Bel602) is a common Arg in the CDR3 α or β (typically the non-germline section), or in the CDR1α (Figure 4, Table 1). This Arg, from different TCR segments, uses its side chain to recognise the peptide, or to bind in between the peptide and the HLA α-helix (Figure 4B). However, this common feature was not associated with higher TCR affinity (Table 1). Strikingly, there is a 10-fold difference in the affinity of the CD4^+^ TCRs restricted to HLA-DQ2/DQ2.2 (average affinity of 39 μM) than the one restricted to HLA-DQ8/8.5 (average affinity of 4 μM). This might be due to the TRBV9*01 gene bias observed in HLA-DQ8^+^ patients, while there is no bias observed in HLA-DQ2^+^ patients [20].

### 2.5. Role of CD4^+^ Th1 Cells in Cancer

Despite the growing number of studies showing an important role of CD4^+^ T cells in cancer [81], the data available on how their TCRs can recognise tumour antigens is limited to two TCRs (E8 and G4 [24,25]) recognising the same human melanoma epitope from the glycolytic enzyme triosephosphate isomerase presented by HLA-DR1 (TPI, Table 1). Strikingly, despite a 100,000-fold increase in the production of GM-CSF by T cells recognising the mutant TPI peptide (Ile28) compared to the wild-type (wt) TPI peptide (Thr28), no detectable affinity for the G4 or E8 TCRs was detected by surface plasmon resonance [25]. The structure of the E8 TCR in complex with both wt and mutant TPI peptide revealed that the increased recognition of the mutant TPI peptide was due to an increase in bonds formed with the CDR3α loop [25]. In 2012, the structure of the G4 TCR in complex with the mutant TPI peptide presented by the HLA-DR1 molecule showed that despite using the same TRAV22 chain, the two TCRs bound with a different footprint [24]. Especially the CDR1α of the E8 and G4 TCRs adopted different conformation, due to differences of length and sequence in the CDR3α loops in the TCRs.

### 2.6. CD4^+^ Th1 Cells with Cytotoxic Properties

Although typically described as helpers, there is mounting evidence that many antiviral IFNγ secreting CD4^+^ T cells can also contribute directly to antiviral immunity and display cytotoxic properties when recognising and responding to peptides presented by HLA-II molecules. Similar to CD8^+^ T cells, cytotoxic CD4^+^ T cells utilise two main cytotoxic pathways; Fas/FasL and Granzyme/Perforin to carry out their cytotoxic mechanisms [13,82]. Protective cytotoxic CD4^+^ T cells have been described in response to a range of viral infections including HIV [9,10], Influenza [11] and HCMV [12]. HIV-specific cytotoxic CD4^+^ T cells are well studied, and were described as early as 2002, where they were detected directly ex vivo and found to be expanded early in HIV infection [9]. Many studies have since expanded upon this, identifying and characterising HIV-specific CD4^+^ T cell responses. Of particular interest are CD4^+^ T cells that recognise the Gag_293_ peptide, which are polyfunctional, cytotoxic, and display a TCRs gene bias for TRAV24/TRBV2. Additionally, a public TCR with high affinity (TRAV24/TRBV2) has been identified in individuals who effectively control HIV infection (HIV controllers) but not in those who cannot [10,83]. Interestingly, despite the critical role of CD4^+^ T cells in HIV infection, it was only in 2018 that a structure of an HIV-specific TCR isolated from a CD4^+^ T cell was solved in complex with the Gag_293_ peptide presented by HLA-II (Table 1). The structure of the public TCR, termed F24 TCR, was solved in complex with the HIV epitope presented by three different HLA-IIs, namely HLA-DR11, -DR1 and -DR15 [10]. The structures revealed how the F24 TCR focused its recognition towards the HIV peptide and conserved parts of the HLA-II molecules (invariant HLA-DRα chain), allowing it to recognise multiple HLA-II molecules.

## 3. T Helper 2 Cells (Th2)

### 3.1. Defining Features of Th2 Cells

Th2 cells were first discovered in mice alongside Th1 cells in 1986 [35], then later in humans, through stimulation of peripheral blood mononuclear cells (PBMCs) with *Toxocara canis* and excretory/secretory antigen [84] (Figure 3). Similarly, activated cells from patients with allergic reactions to timothy grass pollen extract were found to produce IL-4 and IL-5, and low or no IL-2 or IFNγ, and were described as being similar to murine Th2 cells [85]. The primary cytokines secreted by human Th2 cells include IL-4, IL-5, L-13 and, in some cases IL-9 [86]. IL-4 is critical for Th2 differentiation and IL-2 plays a central role in this process [87]. The functions of these cytokines include promoting antibody isotype switching to IgE [88], differentiation, migration and survival of eosinophils [89,90,91], promoting macrophages to adopt an M2 phenotype [92] and the promotion of mucous secretion and fibrosis [93]. Th2 cytokine expression is regulated through the upregulation of the transcription factor GATA3 and the activation of STAT5 along with YY1 [94,95].

### 3.2. Role of Th2 Cells in Atopic Disease

In the normal immune response, Th2 cells play a primary role in combatting infections involving large parasites, such as helminths [96,97]. However, they are also involved in intestinal inflammation and atopic disease [98,99]. Pathogenic Th2 cells (Th2path) are Th2 cells able to secrete IFNγ, IL-17 or TNF, along with the regular Th2 cytokines (Figure 2) [100]. Th2path are memory cells with markers including IL-25, IL-33 and CRT_H_2, although non-pathogenic Th2 cells also express CRT_H_2 but to lower levels [101]. The transcriptional profile of human CRT_H_2^+^ CD4^+^ T cells was studied in the context of asthma, where it was found that *IL17RB,* which encodes for the IL-25R, had a ~3.35-fold increase in transcripts in asthma patients compared to healthy controls [102]. Interestingly, this study found that genes associated with negative regulation of T cell activation were downregulated in Th2 cells from asthma patients compared with healthy individuals [102]. Moreover, it has been found that TCR signalling correlates with the development of asthma [103]. Furthermore, polymorphisms in Differentially Expressed Normal and Neoplastic cells Domain-containing protein 1B (DENND1B) have been associated with asthma and atopic disease in Brazilian children, and DENNI1B knockout in mice has been shown to result in prolonged Th2 signalling and over-secretion of Th2 cytokines [103,104]. DENND1B functions by promoting degradation and internalisation of the TCR, which is necessary during a non-diseased state. This factor functions specifically in Th2 cells in attenuating TCR signalling [103].

Recently, Th2path cells were identified that had the phenotype CD27^-^CD45RB^-^CRT_H_2^+^CD161^+^CD49d^+^ in individuals with numerous allergies, which were absent in individuals with no known allergies [105]. Pathogenic Th2 cells recognise and respond to epitopes which are not pathogen-derived but are from foods, metals or other substances which are not harmful. The superseding CD4^+^ T cell response is known as a hypersensitivity or allergy. Approximately 10–15% of the population suffer from a hypersensitivity to metals; one of the most common hypersensitivities [106,107]. Two Th2 CD4^+^ TCR structures in complex with pHLA-II reveal how T cells can recognise metal ions (nickel and beryllium) involved in acute contact sensitivity [26,27] (Table 1). It was shown that a CD4^+^ TCR from an allergic patient recognises Ni^2+^ bound to an HLA-II molecule containing an unknown self-peptide. Furthermore, mimotope peptides that can replace both the self-peptide and Ni^2+^ in this ligand have been identified and they share a Lysine at position P7 (mimicking the Ni^2+^ ion) that results in the same effect on CD4^+^ T cells as the Ni^2+^ ion. The structure of the Ani2.3 TCR in complex with the HLA-DRB3*03:01 (HLA-DR3) revealed how the non-germline CDR3β loop interacts with the amine group of the P7-Lys side chain [26], which is potentially the binding site of the Ni^2+^ cation. Similar to nickel, beryllium (Be) can also lead to CD4^+^ T cell hypersensitivity in lung inflammatory diseases [108]. The structure revealed that the HLA-DP2 cleft has a large acidic pocket within the antigen binding cleft that favoured the coordination of the Be ion. In 2014, the structure of a Be reactive TCR (AV22) was solved in complex with HLA-DP2 binding the M2 peptide and a Be ion [27]. The structure revealed how the Be ion was buried in the cleft of HLA-DP2. The affinity measurement shows that the AV22 TCR has a high affinity (Kd of 6 μM) for the HLA-DP2-M2 only in the presence of Be. Despite the high affinity of the TCR in the presence of the Be ion, the TCR did not make direct contact with the metal (Figure 4C). Instead, the stable binding of the Be ion changes the surface of the pHLA complex, most likely stabilising the M2 peptide, triggering CD4^+^ T cell recognition and activation [27].

### 3.3. IL-9 Producing Th2 Cells

Pathogenic Th2 cells can secrete high levels of IL-9, a pro-inflammatory cytokine shown to affect different cell types including CD4^+^ T cells, mast cells, B cells, haematopoietic progenitor cells, along with epithelial and smooth muscle cells of the airways [109]. IL-9 producing Th2 CD4^+^ T cells were first identified as being an independent subset of CD4^+^ T cells in mice [110,111] and were referred to as “Th9” cells, which have subsequently been described in multiple reviews which primarily focus on murine research. Prior to this, IL-9 production in mice was associated with CD4^+^ T cells displaying the Th2 phenotype [112]. In 2015, IL-9 producing CD4^+^ T cells were controversially described as an independent subset in humans [113]. Recently, Micosse et al. (2019) identified that IL-9 production was dependent on PPARγ expression, and that all IL-9 producing cells had expression of the Th2 transcription factors GATA3 [114] and PU.1 [115]. Under in vitro conditions, IL-9 producing CD4^+^ T cells, did not display a stable phenotype and when grown under different polarising conditions the cells displayed a Th2 phenotype [114]. Whilst IL-9 producing CD4^+^ T cells do play an important role in human immunity; more research is needed before these cells can be classified as a bona-fide CD4^+^ T cell subset in humans.

IL-9 producing CD4^+^ T cells have been determined to play both a positive and pathogenic role in the adaptive immune response. The population of cells producing IL-9 in humans has been found to primarily consist of skin and gut tropic CD4^+^ T cells [116]. IL-9 producing CD4^+^ T cells have been associated with a number of different diseases including bacterial and fungal skin infections, inflammatory bowel disease (IBD) including ulcerative colitis and Crohn’s disease, autoimmune disorders and allergies affecting the skin including psoriasis and acute contact dermatitis and cancers such as metastatic melanoma [117,118,119]). They have more recently become a major point of interest in the context of IBD due to their role in mediation and progression of the disease [119]. Approximately 41% of individuals with IBD had detectable levels of serum IL-9, and the level of IL-9 in the serum of individuals with Crohn’s disease correlated with disease severity and poor prognosis [120]. However, it should be noted that IL-9 can produced by many other cell types, therefore IL-9 observed in serum does not necessarily originate from IL-9 producing Th2 cells. Despite the involvement of IL-9 producing CD4^+^ T cells in many diseases, no TCR repertoire, and therefore no TCR-pHLA-II structures are available.

## 4. T Helper 17 Cells (Th17)

### 4.1. Defining Features of Th17 Cells

Th17 cells were discovered in 2005 [121] (Figure 3). They produce IL-17A, IL-17F, IL-22, and TNFα making them efficient helpers in eliminating extracellular bacteria and fungi. They preferentially express chemokine receptors CCR6 and CCR4 (CCR6^+^CCR4^+^CXCR3^−^) [122,123]. IL6, IL21, IL23, and TGF-β are the major signalling cytokines involved in Th17 cells differentiation, and the master regulator is retinoic acid receptor-related orphan receptor γ-T (RORγt in humans) [124,125,126]. IL-27 is an important negative regulator of Th17 differentiation [127]. They play an essential role in mucosal immunity in addition to inducing tissue inflammation, Th17 cells also promote B cell responses and the formation of ectopic lymphoid follicles in the tissues [128,129].

Th17 cells are a particularly heterogeneous population with functional states ranging from non-pathogenic to pathogenic, but little is known about these two categories and certain key issues remain to be answered. Non-pathogenic Th17 cells, generated in response to infections with extracellular bacteria [130] are induced by IL-6 and TGF-β1. These cells can produce IL-10 and therefore have significant anti-inflammatory and regulatory properties as well [131,132,133]. Conversely, pathogenic Th17 cells are induced in the presence of IL-6, IL-1β, and IL-23 and can produce pro-inflammatory cytokines and chemokines such as IL-3, IL-22, CXCl3, CCL4 and CCL5 [134].

### 4.2. Role of Th17 Cells in Autoimmune Diseases

Pathogenic Th17 cells have critical roles in the pathogenesis of human autoimmune diseases including RA, Crohn’s disease, IBD, chronic lung inflammation, asthma, psoriasis, and MS [135,136,137]. MS is a chronic inflammatory disease of the central nervous system (CNS). Th17 pathogenicity in MS has been correlated with the dysregulation of the expression of microRNAs (miRNAs), and specific miRNAs promote the pathogenic phenotype [138]. In MS, autoreactive Th17 cells, generated by IL-23 and IL-1β, migrate to the CNS and cross the blood–brain barrier. In encephalitogenic Th17 cells, the Bhlhe40 transcription factor, which is induced by IL-1β signalling, positively regulates the secretion of GM-CSF and consequently, GM-CSF-stimulated glial and dendritic cells reinforce the differentiation and maintenance of pathogenic Th17 cells, by secreting IL-6 and IL-23 [138].

In 2005, the first structures of a TCR bound to HLA-II presenting a myelin basic protein (MBP) derived peptide, which is an autoantigen, were published [28,29] (Table 1, Figure 4D). The first structure showed an MBP derived peptide, MBP_89–101_, presented by HLA-DR2a bound to the 3A6 TCR, revealing that the TCR binds onto the peptide towards the N-terminus, unlike microbial and alloreactive TCRs which bind at the centre of the peptide [29]. The structure of an overlapping MBP peptide, MBP_85–99_, presented by HLA-DR2, was solved bound to the OB.1A12 TCR [28]. The OB.1A12 TCR also contacted only the N-terminus of the peptide, and the interaction was dominated by the CDR3 loops. This binding mode reduces interactions between the TCR and the peptide, and also results in lower binding affinity potentially due to the altered geometry for CD4 coreceptor association [31]. In 2010, another structure was published with the MBP_85–99_ peptide presented by HLA-DQ1, bound to a different TCR (Hy.1B11) [32]. This structure displayed limited interactions between the TCR and the HLA-II due to the unusually small crossing angle of 40° and the strong tilt of 14.5°. The TCR interacted with the peptide using the CDR3α due to the unusually short CDR3β (10 residues long). The remaining MS-related structure was published in 2011 [31] and is of HLA-DR4 presenting a different MBP derived peptide, MBP_114–126_. Interestingly, this MBP peptide is loosely accommodated in the cleft of HLA-DR4 with low affinity (Table 1) to the pHLA, yet the MS2-3C8 TCR has a high affinity of 5.5 µM for the MBP-HLA-DR4 complex. Unlike TCRs 3A6 and OB.1A12, the MS2-3C8 TCR docks onto the MBP-HLA-DR4 complex in a canonical fashion. With the addition of the MS2-3C8- MBP-HLA-DR4 structure, Yin et al. were able to propose two categories of binding topologies for self-reactive TCRs. TCRs that bind in the canonical orientation with structural defects that did not affect their high affinity, and TCRs that bound in altered topologies (ie tilted towards the N terminus) to the pHLA-II and therefore at lower affinity (Figure 4D) [31].

### 4.3. Role of Th17 Cells in Infectious Diseases

Th17 cells have been shown to play a role in the immune response against viral infections in humans including Hepatitis C Virus (HCV), Hepatitis B Virus (HBV) and HIV [136,139,140,141]. In addition, Th17 cells are also required for host defence against intracellular bacterial infection, such as *M. tuberculosis*, *Listeria monocytogenes*, *Chlamydia trachomatis*, *Salmonella enterica*, *Mycoplasma pneumoniae*, *Leishmania donovani*, *Francisella tularensis*, and *Toxoplasma gondii* [142].

Interestingly, bacterial peptides share structural homology with auto-antigens, which might provide a molecular basis for bacterial infection as an onset of autoimmune disease. This is thought to be one of the driving factors of coeliac disease initiation, in which peptides derived from *E. coli* or *Pseudomonas fluorescens* activate Th17 cells that can recognise deamidated gluten by molecular mimicry, resulting in severe inflammation [33]. This is evidenced by three structures of HLA-II presenting a bacterial derived peptide complexed with a TCR (Table 1). All three peptides are unique, and share a common feature permitting molecular mimicry to occur [30,33]. The first structure was HLA-DR2b presenting an E. *coli* derived peptide in complex with the OB.1A12 TCR [30], and the same TCR was initially published bound to HLA-DR2 presenting the MBP_85–99_ peptide [28]. Overlay of the two structures with the MBP_85–99_ peptide [28] and the *E. coli*_346–360_ peptide (GTP-binding protein engA_346–360_) [30] revealed that, despite the lack of sequence homology, the two peptides being presented by different HLA-II molecules were structurally homologous, resulting in molecular mimicry [30]. Importantly, it was noted that the shared sequence motif of P2-His and P3-Phe at the centre of the peptide is considered a hotspot for the OB.1A12 TCR-peptide binding interactions, and this motif along with P1-Val, is shared between both the E. *coli*_346–360_ and MBP_85–99_ peptides. The side chains of these two residues mediate conserved interactions between the OB.1A12 TCR and HLA-II presenting the MBP or E. *coli* peptides [143,144]. Overall, the OB.1A12 TCR docks towards the N-terminus of the E. *coli* peptide-HLA-DR2a complex, and of the MBP-HLA-DR2 structure [28]. The structures of HLA-DQ2-peptide-TCR involving two distinct microbial peptides are mimics of the deamidated gliadin epitopes associated with coeliac disease [33]. Indeed, structural superimposition of HLA-DQ2.5–P. fluor-α1a (bacterial peptide) and of the HLA-DQ2.5-glia-α1 (gliadin peptide) revealed similarity between the both pHLA-II complexes. In addition, the structures of the LS2.8/3.15 TCR-HLA-DQ2.5-P.fluor-α1a [33] and S2 TCR-HLA-DQ2.5-glia-α1 complex [19] show similar distribution of TCR CDR3 loop interactions with the pHLA-II complex [33]. The same observation was made for the structures of HLA-DQ2.5-P. aeru-α2a (bacterial epitope) and HLA-DQ2.5-glia-α2 (gliadin epitope) free or bound with the JR5.1 TCR despite minor variations in docking angle [33].

### 4.4. The Role of Th17 Cells in Other Types of Disease

Th17 cells are also implicated in Graft-versus-host disease (GVHD). In both mice and patients with chronic GVHD, Th17 cell frequencies correlate with disease severity, and strategies that deplete Th17 cells can reverse disease [145]. Moreover, resident memory Th17 cells have been very recently found to be induced by microbial infections in the kidneys and to amplify renal autoimmunity [146].

## 5. T Helper 22 Cells (Th22)

### 5.1. Features of Th22 Cells

Th22 cells were first identified as being a unique subset of CD4^+^ T cells in 2009 [147,148] (Figure 3), and are now classified through the expression of the chemokine receptors CCR6, CCR4 and CCR10, the production of IL-22 and the transcription factors BNC-2, FOXO4, and to a lesser extent AHR [147,148,149] (Figure 2). Expression of CCR10 and IL-22 have been the most common features used to characterise human Th22 cells after activation. Human Th22 cells are tissue-homing CD4^+^ T cells which have been shown to play an important role in skin and gut disease, autoimmunity and allergy [150,151,152]. Skin-tropic Th22 cells localise in higher numbers in the epidermis compared with the dermis, and have also been shown to play a direct role in epidermal wound healing in an IL-22 dependent manner [153]. In culture, human keratinocytes express the IL-22 receptor, which binds IL-22 to increase keratinocyte migration, thickness of the reconstituted epidermis, and downregulation of proteins involved in differentiation [153].

### 5.2. Role of Th22 Cells in the Immune System

The primary disease in which Th22 cells have been studied is RA. Increased IL-22 single-producers were isolated from synovial tissue of patients with RA [154]. In comparison, low levels of infiltration in patients with osteoarthritis were observed compared with patients with RA. The infiltration of Th22 cells was linked with CCL28 secretion, as patients with RA have higher levels of secreted CCL28 than healthy individuals [155]. During RA, Th22 cells act by promoting osteoclast differentiation in an IL-22 dependent manner [155]. Individuals with RA have more Th22 cells and the number of circulating Th22 cells correlates with disease severity, with IL-22 levels increased in patients with RA [150]. Furthermore, skin-tropic Th22 cells were found to be autoreactive to the molecule CD1a [156]. Although there is a structure of a CD8^+^ T cell TCR restricted to CD1a isolated from a patient with SLE [157], no structure has been solved involving a Th22 CD4^+^ TCR.

## 6. Regulatory T Cells (Tregs)

### 6.1. Defining Features of TREGS

The idea of “suppressor T cells” was first established in the 1970′s [158,159] (Figure 3). However, Tregs in humans were not identified until 2001, when multiple groups identified CD4^+^CD25^+^ cells from the thymus and peripheral blood of humans [160,161,162,163]. Tregs account for 1–17% of circulating human CD4^+^ T cells [161,162,163], and are typically identified by the surface expression of the IL-2 receptor α-chain (CD25) [164,165] and the intracellular expression of FoxP3 [166,167,168]. However, the expression of both CD25 and FoxP3, is not restricted to Tregs as CD25 is an activation marker for all T cells and both CD4+ and CD8+ T cells seem to transiently express FoxP3 upon activation [169,170]. As such, the accurate identification of Tregs, particularly in humans, is continually debated. Some studies use additional markers such as CD39, CD28 and CD45RA/RO expression [171,172], or the lack of CD127 expression [173] for further identification and classification. Indeed, a recent study, using mass cytometry, utilised multiple markers to identify a staggering 22 sub-populations of Tregs [174]. To add to the complexity, there are subsets of FoxP3^-^ CD4^+^ T cells found in the periphery, namely Tr1 and Th3 cells, that display Treg like suppressive functions despite being FoxP3 negative [175]. As such, the characterisation of Tregs is a complicated and evolving field of research.

The majority of circulating Tregs are generated in the thymus and are referred to as thymic Tregs (tTregs) or natural Tregs (nTregs) [176]. However, a small subset, referred to as peripheral Tregs (pTregs) develop in the periphery from conventional CD4^+^CD25^-^FoxP3^-^ Tregs if they are activated in the presence of various signals, of particular importance is the presence of transforming growth factor TGF-β which induces the expression of FoxP3 [176,177]. Interestingly, this phenomenon can be manipulated in vitro, and these are referred to as induced/inducible Tregs (iTregs) [176,178,179].

### 6.2. Tregs Are Critical to the Immune System

Tregs are critical to the regulation of the immune system, and their absence leads to autoimmunity. This is highly evident in individuals with IPEX (immune dysregulation, polyendocrinopathy, enteropathy, X-linked) syndrome, where a mutation within FoxP3 [180,181] prevents the generation of Tregs and causes systemic autoimmunity [182]. It is proposed that Tregs manipulate and control immune responses using a variety of mechanisms [183,184,185]. Absent, reduced or suboptimal Treg responses have been implicated in many human autoimmune diseases including Psoriasis [186,187,188,189], MS [190,191,192], Systemic Sclerosis and Morphoea [193,194], SLE [195] and RA [196,197]. Additionally, the role of Tregs have been extensively studied in the development of T1D [198,199,200,201,202,203,204]. Interestingly, the majority of studies demonstrate that individuals suffering with T1D do not have a reduction of Tregs compared to healthy individuals [201,205]; however, there is conflicting evidence about the functional ability of these Tregs [199,200,205]. The mechanisms of Treg dysregulation during T1D are still being uncovered.

In 2015, the first structure of a human CD4^+^ iTreg TCR was solved in complex with a pro-insulin peptide presented by HLA-DR4 [34]. The structure revealed for the first time how a TCR can bind with pHLA in a new and reverse polarity, whereby the TCR adopts a 180° flipped docking (Figure 4E). The reverse topology of the complex, where the α- and β-chains of the TCR are swapped around and dock over the HLA α- and β-chains, imposes constraints for the co-factor binding which would have downstream impacts on T cell signalling. Since this was the first Treg-TCR-pHLA complex and the first TCR shown to adopt a reversed orientation compared to the dogma [1], it was speculated that this unusual topology might be specific to Tregs. However, in 2016, the structure of a naïve murine CD8^+^ TCR was solved showing a reverse docking topology [206]. Furthermore, recent work by Stadinski and colleagues showed that murine Tregs can also dock canonically [207]. Without more Treg-TCR-pHLA structures, it remains unclear if Tregs mostly dock with a reversed polarity, and how this reversed docking impacts on T cell signalling.

### 6.3. Tregs Are an Attractive Target for Immunotherapy

Due to the inducible nature of Tregs in vitro, they are potential targets for immunotherapy, such as targeting of miRNAs to regulate specific Treg populations [208]. Others are trialling the use of low doses of cytokines to selectively enhance Tregs for the treatment of T1D [209] and other autoimmune and inflammatory diseases [210,211]. However, one of the largest areas of research appears to be the use of chimeric receptor antigen (CAR) regulatory T cells [212,213,214]. This involves the ex vivo expansion or in vitro generation of polyclonal [213] and/or epitope-specific [212] CAR Tregs which can subsequently be administered into patients with various autoimmune and inflammatory diseases to enhance their natural Treg levels [215].

### 6.4. The Dark Side of TREGS, How Much Is too Much?

Although Tregs are undoubtedly important in the human immune response, their suppressive nature can actually inhibit immune responses towards some tumours [216]. Tregs can be induced by cytokines and signals present in the tumour microenvironment and as such, can be found in high numbers. This results in an immunosuppressive state, which can reduce the cellular anti-tumour immune response, allowing tumour progression. High numbers of Tregs within the peripheral blood or tumour microenvironment is linked with poor clinical outcomes in some cancers, including Breast cancer [217], Prostate cancer [218], Gastric cancer [219], Lung cancer [220], Renal cancer [221], Ovarian cancer [222], and Cervical cancer [223]. In this instance, it is suggested that Tregs are targeted for immunotherapies to suppress their function, allowing the immune response to infiltrate the tumour microenvironment to enhance cellular anti-cancer immunity.

## 7. T Follicular Helper Cells (Tfh)

### 7.1. Defining Features of Tfh Cells

T follicular helper (Tfh) cells were discovered in 2000, and they contribute to both host protection and inflammatory diseases by providing B-cell help and antibody production [224,225,226] (Figure 3). Tfh cells stay resident in lymph nodes (LNs), in the Germinal Centres (GC) and spleen, and both the upregulation of specific chemokine receptors as well as the direct interaction with B cells in secondary lymphoid organs, are needed for their differentiation [127,132]. Their master regulator is the transcription factor B-cell lymphoma 6 (Bcl-6), and they are also characterized by the expression of the surface receptors CXCR5, ICOS, and PD-1 [227]. In humans, Tfh generation relies on TGF-β, IL-12, IL-23, and Activin A signalling [228,229,230,231,232,233] whereas IL-2 is the most potent inhibitor of their differentiation [234,235]. According to the predominant cytokine secreted, Tfh can be further classified into Tfh1, Tfh2, and Tfh10 cells [126]. Tfh1 secretes IFNγ and promotes IgG2a production; Tfh2 secretes IL-4 and produces IgG1 and IgE; and Tfh10 promotes IgA secretion through the secretion of IL-10 [126]. Tfh cells depend on B cells in most contexts, and GC B and plasma cells depend on Tfh cells. The help provided to GC-B cells by GC-Tfh cells mainly consists of IL-21, IL-4, IL-2, CD40L, CXCL13, and TNF [236].

### 7.2. Role of Tfh Cells in Autoimmune Diseases

Tfh cells have been strongly associated with several autoimmune diseases including; MS [237], SLE [238], RA [236], and T1D [236]). In T1D, it has been found that antigen-specific circulating Tfh (cTfh, in blood) and activated cTfh (PD1^hi^) cells have been observed in children at risk for T1D or with recent onset T1D [236,239,240]. In RA, ectopic lymphoid structures (ELS) in joints, contain B cells, CD4^+^ T cells, and GCs, and are often associated with more severe disease [241]. Tfh cells are implicated in the development or support of ELSs, as Tfh cells produce substantial amounts of CXCL13 in humans, which is required (along with B cells) for ELS development and maintenance [241]. Also, Tfh cells may directly support autoantibody responses by B cells. It has been shown that IL-21-Tfh cells are important in multiple RA animal models, and blocking IL-6 signalling can provide therapeutic benefits in humans with RA [238,242]. Moreover, high numbers of human peripheral blood Tfh cells and enhanced GC formation positively correlates with autoantibody titre and severity of the primary Sjögren’s syndrome [243]. In patients with Autoimmune Myasthenia Gravis, the numbers of cTfh cells are also elevated, and they support autoantibody production, contributing to the development of disease. Tfh cells are also implicated in Ab-mediated allergies, as Tfh cell-derived IL-4 promotes IgE production [228]. Finally, increased frequencies of activated cTfh cells in blood have been reported in SLE patients [238]. Tfh regulates, IL-21 and CD40L, which are both associated with SLE disease [244,245]. One of the most severe manifestations of human SLE is Lupus nephritis, and ectopic GCs and Tfh cells are frequently found in lupus nephritis patient kidneys [236].

### 7.3. Role of Tfh Cells in Infectious Diseases

Tfh cells facilitate Ab responses to bacterial, parasite, fungal, and viral infections. *Streptococcus pyogenes* infection induces an aberrant GC-Tfh cell population with cytotoxic properties (expressing granzyme B) that can directly kill B cells [246]. Tfh cells also play an important role in the control of chronic lymphocytic choriomeningitis (LCMV) infection via the slow development of neutralizing antibodies (nAb) and support of GCs, as well as promoting the secretion of protective Abs for chronic malarial infections [236]. Interestingly, HIV can directly infect Tfh cells [247] and Tfh cells are associated with broad HIV nAb responses [248]. In SIV infection, Env-specific GC-Tfh cells develop and are associated with Env-specific IgG responses [249].

### 7.4. Role of Tfh Cells in Cancer

During the last decade, the role of Tfh cells in cancer has emerged. Tfh-related cells have been found to have immunoprotective functions in breast or colorectal cancer probably via CXCL13 expression [250,251,252]. In tumours, Tfh may help support ELSs, which are a site of recruitment for macrophages, Natural Killer and CD8^+^ T cells, engaging in anti-tumour immunity, or they may promote anti-tumour antibody responses by B cells [253]. Moreover, given that GC-Tfh cells express very high levels of PD1 and are present in lymphoid organs throughout the human body, there can be potential impact of PD1 and PDL1 immunotherapy on Tfh cells [236].

## 8. Conclusions

CD4^+^ T cells assist the immune response in a wide variety of diseases and represent a unique and complex arm of the adaptive immune system, critical for survival. The six distinct CD4^+^ T cell subsets are unique, encompassing different roles and functions. However, it is evident that many of the subsets work in synergy, with some overlap and all have an important role towards infection. Despite the success of the CD4^+^ T cell response, the breakdown of effective regulation can cause significant damage and autoimmunity. To entirely understand and characterise CD4^+^ T cell populations, it is important to consider both the functional and structural characteristics. Currently, there is a lack of diversity of structures solved within the Protein Data Bank with only 14 human allomorphs solved in complex with a TCR, and multiple structures with the same epitope. In order to have a better understanding of the unique molecular mechanisms of the different subsets of CD4^+^ T cells, we hope that future studies will focus on a broader range of epitopes and HLA-II allomorphs, and alongside binding data, we can further define correlates within the subsets that link both the biochemical and functional data. However, by comparing the phenotypes and functions with the known structural data, we have been able to provide detailed insights into the breadth of scope of each of the distinct CD4^+^ T cell lineages. The rapid development of technologies, such as single-cell RNA sequencing, CRISPR genome editing, structural biology (cryo-EM and X-ray crystallography), may lead to the identification of new CD4^+^ T cell subsets. In turn, contributing to a deeper understanding of their molecular requirement, relationship and cross-talks in diseases. Through continued research, we can gain a more thorough understanding about the interactions of CD4^+^ T cells in the broader immune response. This will allow to harness their capabilities to develop new immunotherapies. With exciting new findings in basic CD4^+^ T cell immunology, a deeper understanding will help uncover innovative measures for diseases treatment, thereby improving human health.

## Figures and Tables

**Figure 1 ijms-22-00073-f001:**
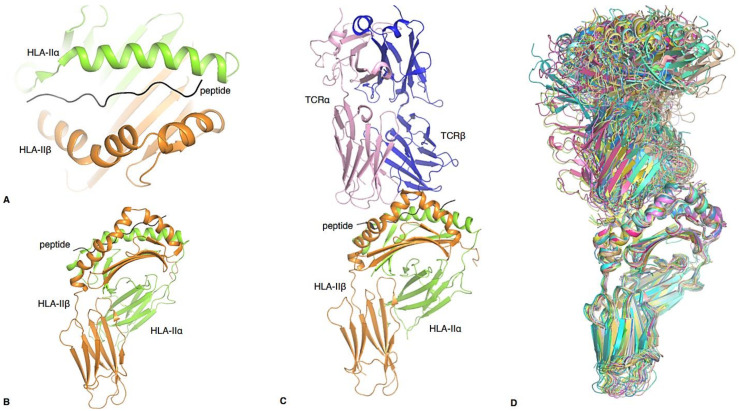
Structural insights into human CD4 TCR recognition. (**A**) Overview of the pHLA-II antigen binding cleft from the top and (**B**) side. The HLA-II α-chain is in orange and the β-chain is in green. The peptide, in black, is bound in the open cleft of HLA-II, with potential overhang residues at each end of the cleft. (**C**) Structure of a CD4 TCR-pHLA-II complex. The F24 TCR is in pink (α-chain) and blue (β-chain), and binding atop the HLA-DR11 (β-chain in orange, HLA-DRα in green) presenting the RQ13 HIV peptide (black). (**D**) Overlay of all known human CD4 TCR-pHLA-II complexes (except GEM42 TCR-CD1b-GMM), further detailed in Table 1.

**Figure 2 ijms-22-00073-f002:**
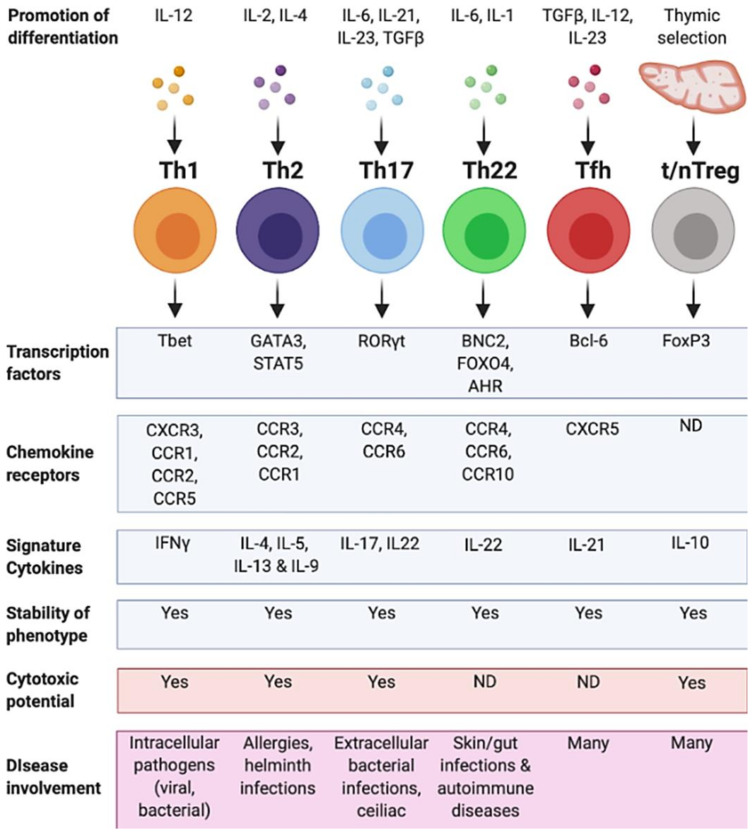
Summary of human CD4^+^ T cell subsets. CD4^+^ T cell helper subsets and their characteristics. Each T helper subtype (Th1, Th2, Th17, Th22, Tfh and thymic/natural (t/n) Treg) is associated with different differentiation agonists, the expression of different transcription factors, chemokine receptors and the secretion of specific cytokines. A CD4^+^ T cell subset also displays a stable phenotype and may have cytotoxic potential. Some subsets have not been defined as having certain characteristics (ND). Each of these characteristics contributes to the cell’s ability to respond in a range of different diseases. Figure created using Biorender.com.

**Figure 3 ijms-22-00073-f003:**
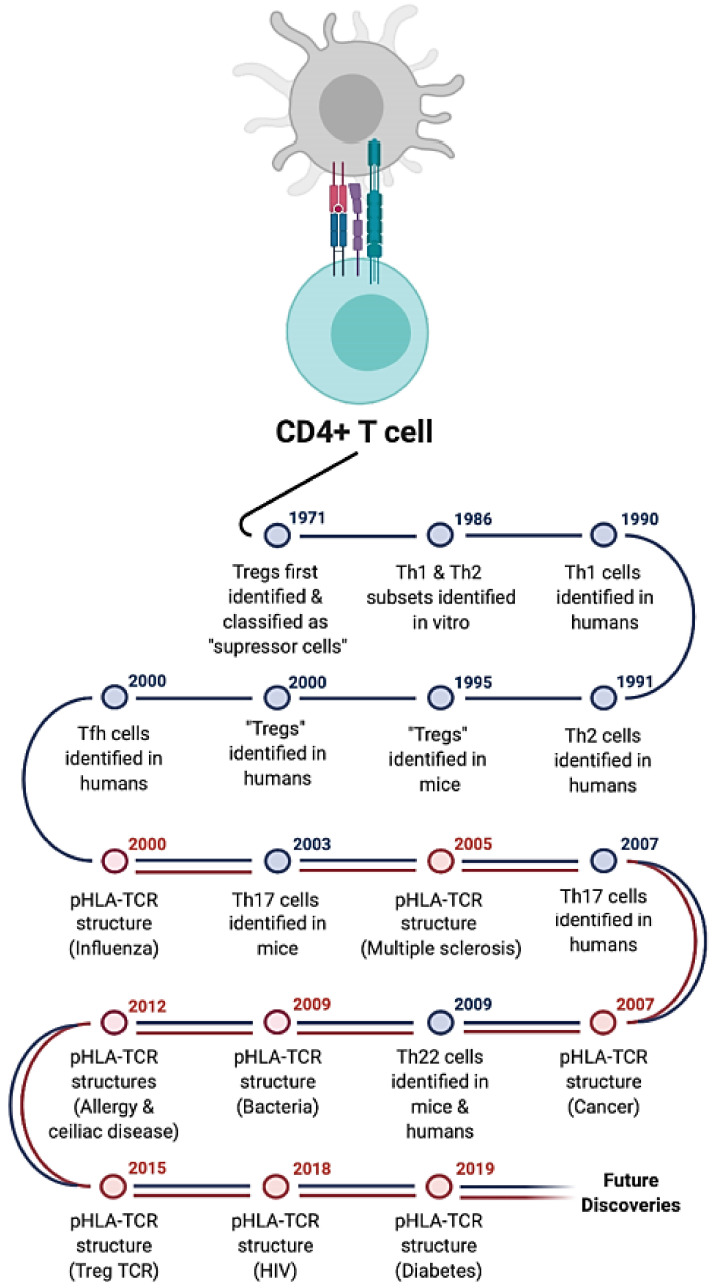
Timeline for the discovery of CD4^+^ T cell subsets. Timeline of CD4^+^ T cell subset discovery and major pHLA-TCR structures. Blue timepoints indicate major immunological discoveries in the identification of the CD4^+^ T cell subsets and red timepoints indicate the publication of major pHLA-TCR structures. The disease from which the peptide was derived or the type of helper subset of the TCR is written in parentheses. Figure created using Biorender.com.

**Figure 4 ijms-22-00073-f004:**
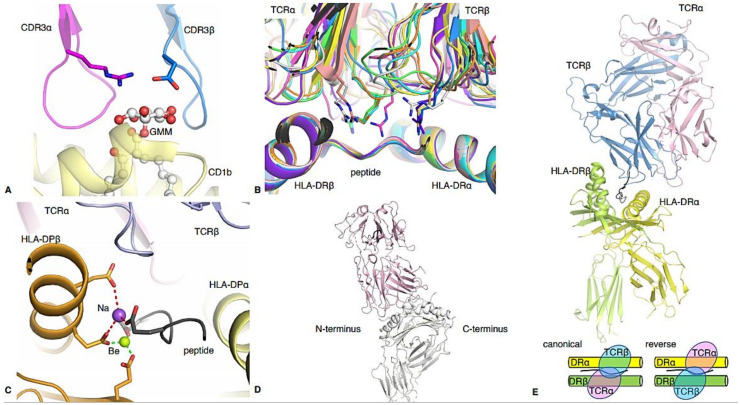
Structural highlights of some human CD4 TCR-pHLA-II complexes. (**A**) GEM42 TCR CDR3α loop in pink, and CDR3β loop in blue, wrapping the GMM lipid-based antigen (white stick and spheres) presented by the HLA-like molecule CD1b (yellow). (**B**) Structural overlay of all gluten-specific TCRs bound to their antigen, alignment made on the antigen binding cleft. The molecules are represented as cartoons, and the characteristic Arginine residue from TCRs is shown as a stick. The complexes are coloured in green (PDB code: 4Z7U), cyan (PDB code: 4Z7V), pink (PDB code: 4Z7W), yellow (PDB code: 5K9S), salmon (PDB code: 5KSA), white (PDB code: 5KSB), purple (PDB code: 4OZH), pale yellow (PDB code: 4OZG), blue (PDB code: 4OZF) and orange (PDB code: 4GG6). (**C**) AV22 TCR (α-chain in pink and β-chain in blue) binding to the HLA-DP2 molecule (α-chain in yellow and β-chain in orange) complexed with the mimotope M2 peptide (grey) and Be (green sphere) and Na (purple sphere). The bonds formed between the pHLA and the Na or Be are represented by red or green dashed lines, respectively. (**D**) Representative docking topology of the majority of CD4 Th17 TCRs binding towards the N-terminus of the pHLA, represented by the 3A6 TCR in light pink [29]. (**E**) CD4 Treg FS18 TCR in complex with the proinsulin-HLA-DR4 (HLA-DRβ chain in green, HLA-DRα in yellow, peptide in black). The TCRα chain is in pink and over the HLA-DRα chain, while the TCRβ chain is coloured in blue and docks over the HLA-DRβ chain, exhibiting a reversed topology. The bottom panel show a schematic of canonical and reversed docking topology for αβTCRs, respectively.

**Table 1 ijms-22-00073-t001:** List of human CD4 TCR-antigen-HLA or HLA-like complex structure.

HLA or CD1	Peptide Sequence	TCR	TRAV	TRBV	K_d_ (μM)	Role in Disease	PBD	Reference
HLA-DR1	PKYVKQNTLKLAT	F11	8-4*01	24-1*01	26.7	Antiviral	6R0E	[15]
HLA-DR1	PKYVKQNTLKLAT	HA1.7	8-4*01	28*01	37	Antiviral	1FYT	[16]
HLA-DR1	RFYKTLRAEQAS	F24	24*01	2*01	10.56	Antiviral	6CQR	[10]
HLA-DR11	RFYKTLRAEQAS	F24	24*01	2*01	1.16	Antiviral	6CQL	[10]
HLA-DR15	RFYKTLRAEQAS	F24	24*01	2*01	6.9	Antiviral	6CQQ	[10]
CD1b	GMM-C32	GEM42	1-2*01	6-2*01	0.85	Anti-bacterial	5L2K	[17]
HLA-DQ8	VEELYLVAGEEGC	T1D-3	17*01	5-1*01	21.4	Diabetes	6DFX	[18]
HLA-DQ2	PQPELPYPQ	D2	26-1*01	7-2*01	15.8	Coeliac disease	4OZH	[19]
HLA-DQ2	PQPELPYPQ	JR5.1	26-1*01	7-2*01	79.4	Coeliac disease	4OZG	[19]
HLA-DQ2	PQPELPYPQ	S16	26-1*01	7-2*01	24.8	Coeliac disease	4OZF	[19]
HLA-DQ2	PFPQPELPY	S2	4*01	20-1*01	70	Coeliac disease	4OZI	[19]
HLA-DQ2.2	PFSEQEQPV	T1005.2.56	21*01	7-3*01	22.1	Coeliac disease	6PX6	[20]
HLA-DQ2.2	PFSEQEQPV	T594	9-2*01	11-2*01	20.9	Coeliac disease	6PY2	[20]
HLA-DQ8	SGEGSFQPSQENP	S13	26-2*01	9*01	1.05	Coeliac disease	4Z7U	[21]
HLA-DQ8	SGEGSFQPSQENP	L3-12	26-2*01	9*01	7	Coeliac disease	4Z7V	[21]
HLA-DQ8	SGEGSFQPSQENP	T316	8-3*01	6-1*01	2.1	Coeliac disease	4Z7W	[21]
HLA-DQ8	SGEGSFQPSQENP	Bel502	20*01	9*01	2.8	Coeliac disease	5KS9	[22]
HLA-DQ8	QPQQSFPEQEA	Bel602	20*01	9*01	4.7	Coeliac disease	5KSA	[22]
HLA-DQ8	SGEGSFQPSQENP	SP3.4	26-2*01	9*01	11.4	Coeliac disease	4GG6	[23]
HLA-DQ8.5	GPQQSFPEQEA	T15	20*01	9*01	2	Coeliac disease	5KSB	[22]
HLA-DR1	GELIGILNAAKVPAD	G4	22*01	5-8*01	low ^$^	Cancer	4E41	[24]
HLA-DR1	GELIGILNAAKVPAD	E8	22*01	6-6*01	low ^$^	Cancer	2IAM	[25]
HLA-DR3 ^#^	QHIRCNIPKRISA	ANi2.3	8-3*01	19*01	38.3	Allergy	4H1L	[26]
HLA-DP2	QAFWIDLFETIG	AV22	9-2*01	5-1*01	6	Allergy	4P4K	[27]
HLA-DR2	ENPVVHFFKNIVTP	OB.1A12	17*01	20-1*01	47	Multiple sclerosis	1YMM	[28]
HLA-DR2a	VHFFKNIVTPRTPGG	3A6	9-2*02	5-1*01	low ^$^	Multiple sclerosis	1ZGL	[29]
HLA-DR2b	DFARVHFISALHGSG	OB.1A12	17*01	20-1*01	300	Anti-bacterial	2WBJ	[30]
HLA-DR4	FSWGAEGQRPGFG	MS2-3C8	26-2*01	20-1*01	5.5	Multiple sclerosis	3O6F	[31]
HLA-DQ1	ENPVVHFFKNIVTPR	Hy.1B11	13-1*02	7-3*01	14.3	Multiple sclerosis	3PL6	[32]
HLA-DQ2	APMPMPELPYP	LS2.8/3.15	8-3*01	5-5*01	39.6	Anti-bacterial	6U3N	[33]
HLA-DQ2	AVVQSELPYPEGS	JR5.1	26-1*01	7-2*01	132	Anti-bacterial	6U3O	[33]
HLA-DR4	GSLQPLALEGSLQKRGIV	FS18	29/DV5*01	6-2*01	low ^$^	Treg	4Y19	[34]

The α and β gene usage are reported accordingly to the IMGT nomenclature, with TRAV for TCR α variable domain and TRBV for the TCR β variable domain segments. The different colours highlight the CD4^+^ T cell subset within which the TCR fits accordingly to their role in immunity, with Th1 like responses in green, Th2 like responses in yellow, Th17 like responses in orange, and blue for the Treg like response. ^#^ HLA-DR3 is short for HLA-DRB3*03:01. ^$^ low refers to surface plasmon resonance performed for which the results show either no binding using a standard protocol (monomer) or very weak binding at a high concentration of analyte.

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
