# Peer review of "The Many Faces of CD4+ T Cells: Immunological and Structural Characteristics"

_ijms, 2020, doi:10.3390/ijms22010073_

Round 1

Reviewer 1 Report

In this ambitious (because of its scope) and well written review, the authors describe the various human subsets of CD4 T helper cells. The manuscript is informative, provides a good historical perspective, and is original by including rather detailed structural information of TCRs used by the different Th subsets. A main weakness which can be easily corrected is the excessive usage of reviews (rather than primary research papers) in the reference list.

Specific comments:

  1. The manuscript focuses on human (not murine) Th cells. It should be indicated in the Abstract, and possibly in the title as well.
  2. All statements in a review should be supported by one or several references pointing to the original (first) report(s). This is important both to give credits to the scientist(s) who made the key findings and to provide a correct historical perspective. Accordingly, all references should be carefully selected and contain data that support each statement made in the review. A few examples of missing or inappropriate references are listed below but this remark applies for the whole manuscript.
  3. Page 2, lines 47-48. A reference to a primary research paper would be appropriate after:“ CD4+ T cells license dendritic cells to assist in the activation of naive CD8+ T cells, which are the killers of the immune system.”
  1. Page 2, lines 48-49. Similarly, a reference to a primary research paper would be appropriate after: “Furthermore, CD4+ T cells help B cells, which predominately make antibodies.”
  2. In the Introduction, it could be mentioned that Th1 cells help macrophages to kill intracellular bacteria and cancer cells. (IFNgamma was first known as “macrophage activating factor”).
  3. Figure 2. At the top, IL-2 and IL-23 are listed for “promotion of differentiation” of Th2 cells, which is quite surprising and would need to be explained in the main text. IL-4 (and IL-4 only) is generally considered the polarizing cytokine for Th2 differentiation.
  4. Page 2, lines 57-58, and Figure 2. It is stated that Th subsets have “a relatively stable phenotypic profile”. I would agree with this statement. However, the plasticity of Th subsets has received enormous attention is recent years. Accordingly, the current mainstream view seems to be that Th subsets have a plastic rather than a stable phenotypic profile. Some scientists go as far as saying that (stable) Th subsets do not exist in humans. Therefore, I would suggest the authors to put forward some arguments why they think that Th subset stability is the rule and plasticity the exception.
  1. Figure 2 and main text. Th1 cells typically secrete TNFalpha as well, don’t they?
  2. Figure 3 and main text. The statement “Tregs first identified as suppressor cells” is a bit problematic because T suppressor cells were known as expressing the CD8 co-receptor, I believe.
  3. Figure 3. References could be included in the timeline.
  4. Figure 4. Where do the structures shown come from? References to published papers would be appropriate.
  5. Page 9, lines 169-171: In “In the absence of effective regulation, Th1 CD4+ T cell responses can target self-peptides, resulting in autoimmune diseases such as Type 1 diabetes (T1D) [44], Rheumatoid Arthritis (RA) [45], and Systemic Lupus Erythematosus (SLE) [45].”, reference #45 (and probably #44 also) is clearly inappropriate. (See point #3 above)
  6. Page 10, lines 187-188 (and other places): “Coeliac disease is another autoimmune disease caused by circulating Th1 CD4+ T cells which respond to gliadin derived peptides from oral gluten”. It should be made clear for the readers that gluten and gliadin-derived peptides are food-derived foreign antigens and not self antigens!
  7. Section 2.6 (CD4+ Th1 cells with cytotoxic properties). Please explain if killing of target cells by cytotoxic Th1 cells is associated with antigen presentation by HLA-II, in a similar fashion as for cytotoxic CD8 T cell killing target cells expressing specific antigen presented by HLA-I. If so, which cells can be killed by cytotoxic Th1 cells? Most cells normally do not express HLA-II. Please clarify.
  8. Page 11, lines 252-253, a reference is needed (preferably to a primary research paper) after “In the normal immune response, Th2 cells play a primary role in combatting infections involving large parasites, such as helminths.”
  9. Section 3.2 The part on T-cell “recognition” of metal ions is very interesting, but the structural aspects could be explained even better (possibly with an additional figure). As it is written now, one may think that metal ions can take the place and be recognized by the TCR instead of a peptide in the HLA-II groove. Please clarify.
  10. Bottom of page 12. It should be mentioned that IL-9 is produced by many other cell types that Th2 cells, and that IL-9 detected in serum does not necessarily originate from IL-9 producing Th2 cells.
  11. What do the authors think about Th cells that secrete both IFNgamma and IL-17? These cells are often referred to as IFNgamma-producing Th17 cells, but they could as well be classified as IL-17-producing Th1 cells, I guess. Please clarify.
  12. Section 6.1 (Defining features of Tregs). It should be made clear for the readers that CD25 is an activation marker for all T cells. CD25 expression is not restricted to Treg cells. CD25 is the receptor for IL-2 which is arguably the most important growth factor for all T cells.
  13. Section 6.1 (Defining features of Tregs). It should be clearly mentioned that most human CD4+ and CD8+ T cells seem to transiently express Foxp3 upon activation (PMID: 17329235, 15620457, 15902688, 17154262, 17644734, 17185041). Thus, Foxp3 expression is not strictly restricted to Treg cells.
  14. Page 16, lines 487-488. The statement “High numbers of Tregs within the peripheral blood or tumour microenvironment is linked with poor clinical outcomes in several cancers including…” needs to me moderated because high densities of Foxp3+ cells in tumor are associated with good prognosis for several types of cancers such as colorectal cancer (PMID: 29088871) and lymphoma (PMID: 18223287).
  15. Most references in a review (typically 90% or more) should point to original primary research papers rather than previously published reviews. A review of reviews is not particularly useful to the readers. A few exceptions can be made for outstanding reviews and opinion/conceptual articles. Two examples are listed below but this remark applies for the whole manuscript. I would recommend the authors to replace most reviews in the reference list with appropriate primary research papers.
  16. Page 11, lines 222-223. After “Similar to CD8+ T cells, cytotoxic CD4+ T cells utilise two main cytotoxic pathways; Fas/FasL and Granzyme/Perforin to carry out their cytotoxic mechanisms [7].”, a reference to one or several primary research paper(s) would be much better than to a review.
  17. Page 13, lines 322-324. After “IL6, IL21, IL23, and TGF-beta are the major signalling cytokines involved in Th17 cells differentiation, and the master regulator is retinoic acid receptor-related orphan receptor g-T (ROR t in humans) [103].”, a reference to a primary research paper would be much better than to a review.

Author Response

Chatzileontiadou et al Point by Point response. 

We would like to thank the three reviewers for their careful review of our manuscript, their positive comments, and constructive remarks that we have addressed and believe make our manuscript stronger. We provide below a point by point reply to each reviewer’s comments, and we have updated our manuscript and figures accordingly. 

Reviewer #1

Comments and Suggestions for Authors

In this ambitious (because of its scope) and well written review, the authors describe the various human subsets of CD4 T helper cells. The manuscript is informative, provides a good historical perspective, and is original by including rather detailed structural information of TCRs used by the different Th subsets. A main weakness which can be easily corrected is the excessive usage of reviews (rather than primary research papers) in the reference list.

We thank Reviewer #1 for the careful assessment of our review and for their positive comments. Since this special issue of IJMS focuses on the advances in the field of T cell immunity over the last 10 years, we chose to limit some older primary research papers and instead reference some recent primary research papers or elegant and detailed reviews. Although these are not the first reports, they are appropriate for the points they are referencing.

Specific comments:

1. The manuscript focuses on human (not murine) Th cells. It should be indicated in the Abstract, and possibly in the title as well.

We have now included a reference to humans in the Abstract (Page 1, Line 19) to read

“In this review, we discuss the defining features of the main human CD4+ T cell lineages and their role in immunity, as well as the structural characteristics underlying their detection of pathogens.”

2. All statements in a review should be supported by one or several references pointing to the original (first) report(s). This is important both to give credits to the scientist(s) who made the key findings and to provide a correct historical perspective. Accordingly, all references should be carefully selected and contain data that support each statement made in the review. A few examples of missing or inappropriate references are listed below but this remark applies for the whole manuscript.

As per our answer above, this Special Issue of IJMS focuses on the advances in the field of T cell immunity over the last 10 years and as such we limited older primary research papers in favour of carefully selected recent primary recent research articles or reviews that thoroughly detail the relevant area. None-the-less, we have now included some additional primary research papers, detailed below.

3. Page 2, lines 47-48. A reference to a primary research paper would be appropriate after: “ CD4+ T cells license dendritic cells to assist in the activation of naive CD8+ T cells, which are the killers of the immune system.”

We have added the following primary research paper as a reference (Page 2, Line 50):

[7] Ridge JP, Di Rosa F, Matzinger P. A conditioned dendritic cell can be a temporal bridge between a CD4+ T-helper and a T-killer cell. Nature. 1998;393(6684):474-8.

4. Page 2, lines 48-49. Similarly, a reference to a primary research paper would be appropriate after: “Furthermore, CD4+ T cells help B cells, which predominately make antibodies.”

We have added the following primary research article as a reference (Page 2, Line 51):

[8] Claman HN, Chaperon EA, Triplett RF. Thymus-marrow cell combinations. Synergism in antibody production. Proc Soc Exp Biol Med. 1966;122(4):1167-71.

5. In the Introduction, it could be mentioned that Th1 cells help macrophages to kill intracellular bacteria and cancer cells. (IFNgamma was first known as “macrophage activating factor”).

Since the introduction is quite broad, we have instead included this in the Th1 section with new references which now reads (Page 8, Lines 100-102): 

“IFNg, which was first known as macrophage-activating factor [37] (Figure 2), playing a central role in macrophage activation against intracellular bacteria and cancer cells [38, 39].

[37] Pace JL, Russell SW, Schreiber RD, Altman A, Katz DH. Macrophage activation: priming activity from a T-cell hybridoma is attributable to interferon-gamma. Proc Natl Acad Sci U S A. 1983;80(12):3782-6.

[38] Keller R, Fischer W, Keist R, Bassetti S. Macrophage response to bacteria: induction of marked secretory and cellular activities by lipoteichoic acids. Infection and immunity. 1992;60(9):3664-72.

[39] Muller E, Christopoulos PF, Halder S, Lunde A, Beraki K, Speth M, Oynebraten I, Corthay A. Toll-Like Receptor Ligands and Interferon-gamma Synergize for Induction of Antitumor M1 Macrophages. Frontiers in immunology. 2017;8:1383.

6. Figure 2. At the top, IL-2 and IL-23 are listed for “promotion of differentiation” of Th2 cells, which is quite surprising and would need to be explained in the main text. IL-4 (and IL-4 only) is generally considered the polarizing cytokine for Th2 differentiation.

We thank the reviewer for this correction and we have now removed IL-23 from the text and figure. We have however added the following reference regarding the role of IL-2 in Th2 cell differentiation, which now reads (Page 11, line 252-253):

 “IL-4 is critical for Th2 differentiation and IL-2 plays a central role in this process [87].”

[87] Cote-Sierra J, Foucras G, Guo L, Chiodetti L, Young HA, Hu-Li J, Zhu J, Paul WE. Interleukin 2 plays a central role in Th2 differentiation. Proc Natl Acad Sci U S A. 2004;101(11):3880-5.

7. Page 2, lines 57-58, and Figure 2. It is stated that Th subsets have “a relatively stable phenotypic profile”. I would agree with this statement. However, the plasticity of Th subsets has received enormous attention is recent years. Accordingly, the current mainstream view seems to be that Th subsets have a plastic rather than a stable phenotypic profile. Some scientists go as far as saying that (stable) Th subsets do not exist in humans. Therefore, I would suggest the authors to put forward some arguments why they think that Th subset stability is the rule and plasticity the exception.

We thank the reviewer for raising this critical point. Indeed, there is much debate about the plasticity or stability of these Th subsets, the details of which are outside of the scope of this review since we focus on the main characteristics of each subset and their role in disease. To remove any potential for confusion, we have rephrased the following sentence to read (Page 2, Line 60):

 “This facilitates the differentiation of CD4+ T cells into a range of subsets characterised by unique cytokine profiles, specific transcription factor expression, and a relatively distinct phenotypic profile (Figure 2)”

8. Figure 2 and main text. Th1 cells typically secrete TNFalpha as well, don’t they?

We agree with the reviewer and have added this the overview of Th1 cells to read (Page 8, Line 108):

“Additionally, Th1 CD4+ T cells may also produce IL-2 [35, 48], IL-10 [49, 50], TNF-α [51] and lymphotoxin [35]

[51] Austin LM, Ozawa M, Kikuchi T, Walters IB, Krueger JG. The majority of epidermal T cells in Psoriasis vulgaris lesions can produce type 1 cytokines, interferon-gamma, interleukin-2, and tumor necrosis factor-alpha, defining TC1 (cytotoxic T lymphocyte) and TH1 effector populations: a type 1 differentiation bias is also measured in circulating blood T cells in psoriatic patients. J Invest Dermatol. 1999;113(5):752-9.

9. Figure 3 and main text. The statement “Tregs first identified as suppressor cells” is a bit problematic because T suppressor cells were known as expressing the CD8 co-receptor, I believe.

In Section 6.1 – Defining features of Tregs, we indicate that suppressor cells were first identified in the 1970s, the term used by Taussig et al [159], but further indicate that bona fide Tregs were not identified in humans until the 2000s. To remove any potential confusion around this point, we have put the term “suppressor T cells” in quotation marks to suggest they are a concept more than a defined subset, to now read (Page 15, Line 441):   

“The idea of “suppressor T cells” was first established in the 1970’s [158, 159]”.

10.Figure 3. References could be included in the timeline.

References are now included in Figure 3

11. Figure 4. Where do the structures shown come from? References to published papers would be appropriate.

To make it clear, we have added the words “PDB code” in Figure 4’s legend (Page 9, Lines 160-162) to clarify that the codes are from an official published database. The relevant published references for all structures mentioned in the review are included in Table 1.

12. Page 9, lines 169-171: In “In the absence of effective regulation, Th1 CD4+ T cell responses can target self-peptides, resulting in autoimmune diseases such as Type 1 diabetes (T1D) [44], Rheumatoid Arthritis (RA) [45], and Systemic Lupus Erythematosus (SLE) [45].”, reference #45 (and probably #44 also) is clearly inappropriate. (See point #3 above)

Both references are recent primary research articles. Although they are not the first reports, they are relevant and very recent references, keeping with the theme of this special issue edition.

13. Page 10, lines 187-188 (and other places): “Coeliac disease is another autoimmune disease caused by circulating Th1 CD4+ T cells which respond to gliadin derived peptides from oral gluten”. It should be made clear for the readers that gluten and gliadin-derived peptides are food-derived foreign antigens and not self antigens!

We have clarified this to now read (Page 10, Line 193):

“Coeliac disease is another autoimmune disease caused by circulating Th1 CD4+ T cells which respond to gliadin derived peptides from oral food-derived gluten, and induce inflammation within the mucosa of the gastrointestinal tract [77, 78].”

14. Section 2.6 (CD4+ Th1 cells with cytotoxic properties). Please explain if killing of target cells by cytotoxic Th1 cells is associated with antigen presentation by HLA-II, in a similar fashion as for cytotoxic CD8 T cell killing target cells expressing specific antigen presented by HLA-I. If so, which cells can be killed by cytotoxic Th1 cells? Most cells normally do not express HLA-II. Please clarify.

In this section we are referring to Th1 cells with the added ability to undertake cytotoxic functions and therefore are referring solely to their recognition of peptides presented by HLA-II. Cytotoxic Th1 cells have been shown to recognise peptides presented by HLA-II in the context of multiple viral infections, and have been particularly well studied in the context of HIV and influenza infections. To clarify this point, we have added the following sentence into the text to now read (Page 11, Line 227):

“Although typically described as helpers, there is mounting evidence that many antiviral IFNg secreting CD4+ T cells can also contribute directly to antiviral immunity and display cytotoxic properties when recognising and responding to peptides presented by HLA-II molecules”

HLA-II molecules are present on the surface of many different cell types, any of which could be killed by cytotoxic CD4+ T cells. The expression of HLA-II on different cell types is important for all CD4+ T cell subsets, and so we have added more details in the introduction to now read (Page 1, Lines 30-31):

“Following selection and maturation in the thymus, naïve CD4+ T cells migrate to the periphery where they survey for antigens displayed by human leukocyte antigen class II (HLA-II) molecules present on the surface of professional antigen presenting cells (APCs) [1] such as Dendritic cells (DCs) [2], B cells [3], macrophages [4], CD4+ T cells and airway and intestinal epithelial cells [5]

[2] Ohno Y, Kitamura H, Takahashi N, Ohtake J, Kaneumi S, Sumida K, Homma S, Kawamura H, Minagawa N, Shibasaki S, et al. IL-6 down-regulates HLA class II expression and IL-12 production of human dendritic cells to impair activation of antigen-specific CD4(+) T cells. Cancer immunology, immunotherapy : CII. 2016;65(2):193-204.

[3] Rijvers L, Melief MJ, van Langelaar J, van der Vuurst de Vries RM, Wierenga-Wolf AF, Koetzier SC, Priatel JJ, Jorritsma T, van Ham SM, Hintzen RQ, et al. The Role of Autoimmunity-Related Gene CLEC16A in the B Cell Receptor-Mediated HLA Class II Pathway. J Immunol. 2020;205(4):945-56.

[4] Codolo G, Toffoletto M, Chemello F, Coletta S, Soler Teixidor G, Battaggia G, Munari G, Fassan M, Cagnin S, de Bernard M. Helicobacter pylori Dampens HLA-II Expression on Macrophages via the Up-Regulation of miRNAs Targeting CIITA. Frontiers in immunology. 2019;10:2923.

[5] Wosen JE, Mukhopadhyay D, Macaubas C, Mellins ED. Epithelial MHC Class II Expression and Its Role in Antigen Presentation in the Gastrointestinal and Respiratory Tracts. Frontiers in immunology. 2018;9:2144.

15. Page 11, lines 252-253, a reference is needed (preferably to a primary research paper) after “In the normal immune response, Th2 cells play a primary role in combatting infections involving large parasites, such as helminths.”

We have added the following references (Page 11, Line 260):

[96] Allen JE, Sutherland TE. Host protective roles of type 2 immunity: parasite killing and tissue repair, flip sides of the same coin. Semin Immunol. 2014;26(4):329-40.

[97] Zaiss MM, Maslowski KM, Mosconi I, Guenat N, Marsland BJ, Harris NL. IL-1beta suppresses innate IL-25 and IL-33 production and maintains helminth chronicity. PLoS pathogens. 2013;9(8):e1003531.

16. Section 3.2 The part on T-cell “recognition” of metal ions is very interesting, but the structural aspects could be explained even better (possibly with an additional figure). As it is written now, one may think that metal ions can take the place and be recognized by the TCR instead of a peptide in the HLA-II groove. Please clarify.

We thank the reviewer for pointing this out and have now clarified the text to read (Page 12, Lines 283-287):

 “It was shown that a CD4+ TCR from an allergic patient recognises Ni2+ bound to an HLA-II molecule containing an unknown self-peptide. Furthermore, mimotope peptides that can replace both the self-peptide and Ni2+ in this ligand have been identified and they share a Lysine at position P7 (mimicking the Ni2+ ion) that results in the same effect on CD4+ T cells as the Ni2+ ion.”

17. Bottom of page 12. It should be mentioned that IL-9 is produced by many other cell types that Th2 cells, and that IL-9 detected in serum does not necessarily originate from IL-9 producing Th2 cells.

We have added the following sentence (Page 13, Line 325-326):

“However, it should be noted that IL-9 can produced by many other cell types, therefore IL-9 observed in serum does not necessarily originate from IL-9 producing Th2 cells.”

18. What do the authors think about Th cells that secrete both IFNgamma and IL-17? These cells are often referred to as IFNgamma-producing Th17 cells, but they could as well be classified as IL-17-producing Th1 cells, I guess. Please clarify.

We agree with the reviewer that some of these cytokines can be produced by multiple subsets. Indeed, the classification of CD4+ T cell subsets is complicated and at times controversial. To maintain clarity within our review we have been very deliberate and clear on our definition of a “subset” to prevent any confusion. Typically, cells producing both IL-17 and IFNg are classified as Th17 cells, as indicated by the reviewer. As such, we have also classified them as Th17 cells and feel a more in-depth discussion around this may be confusing to readers, and is therefore outside the scope of this review.

19. Section 6.1 (Defining features of Tregs). It should be made clear for the readers that CD25 is an activation marker for all T cells. CD25 expression is not restricted to Treg cells. CD25 is the receptor for IL-2 which is arguably the most important growth factor for all T cells.

We have included this important point with the following sentence (Page 15, lines 446-448):

“However, the expression of both CD25 and Foxp3, is not restricted to Tregs as CD25 is an activation marker for all T cells and both CD4+ and CD8+ T cells seem to transiently express FoxP3 upon activation [169, 170]. As such, the accurate identification of Tregs, particularly in humans, is continually debated.”

[169] Roncador G, Brown PJ, Maestre L, Hue S, Martinez-Torrecuadrada JL, Ling KL, Pratap S, Toms C, Fox BC, Cerundolo V, et al. Analysis of FOXP3 protein expression in human CD4+CD25+ regulatory T cells at the single-cell level. Eur J Immunol. 2005;35(6):1681-91.

[170] Pillai V, Ortega SB, Wang CK, Karandikar NJ. Transient regulatory T-cells: a state attained by all activated human T-cells. Clinical immunology. 2007;123(1):18-29.

20. Section 6.1 (Defining features of Tregs). It should be clearly mentioned that most human CD4+ and CD8+ T cells seem to transiently express Foxp3 upon activation (PMID: 17329235, 15620457, 15902688, 17154262, 17644734, 17185041). Thus, Foxp3 expression is not strictly restricted to Treg cells.

This has been included above (Comment 19).

21. Page 16, lines 487-488. The statement “High numbers of Tregs within the peripheral blood or tumour microenvironment is linked with poor clinical outcomes in several cancers including…” needs to me moderated because high densities of Foxp3+ cells in tumor are associated with good prognosis for several types of cancers such as colorectal cancer (PMID: 29088871) and lymphoma (PMID: 18223287).

Since this section discusses the negative effects of Tregs, we have chosen to focus on cancers where the presence of elevated Tregs are associated with poor clinical outcomes. We have subsequently removed colorectal cancer from the sentence due to conflicting reports of the role of Tregs in clinical outcomes. We have also amended a few words to clarify to readers that this is not relevant to all cancers. The section now reads (Page 16, Lines 497-503): 

“Although Tregs are undoubtedly important in the human immune response, their suppressive nature can actually inhibit immune responses towards some tumours [216]. Tregs can be induced by cytokines and signals present in the tumour microenvironment and as such, can be found in high numbers. This results in an immunosuppressive state, which can reduce the cellular anti-tumour immune response, allowing tumour progression. High numbers of Tregs within the peripheral blood or tumour microenvironment is linked with poor clinical outcomes in some cancers including; Breast cancer [217], Prostate cancer [218], Gastric cancer [219], Lung cancer [220], Renal cancer [221], Ovarian cancer [222], and Cervical cancer [223].”

References for Gastric cancer [219], Lung cancer [220] and Renal cancer [221] have been corrected.

[219] Perrone G, Ruffini PA, Catalano V, Spino C, Santini D, Muretto P, Spoto C, Zingaretti C, Sisti V, Alessandroni P, et al. Intratumoural FOXP3-positive regulatory T cells are associated with adverse prognosis in radically resected gastric cancer. European journal of cancer. 2008;44(13):1875-82.

[220] O'Callaghan DS, Rexhepaj E, Gately K, Coate L, Delaney D, O'Donnell DM, Kay E, O'Connell F, Gallagher WM, O'Byrne KJ. Tumour islet Foxp3+ T-cell infiltration predicts poor outcome in nonsmall cell lung cancer. The European respiratory journal. 2015;46(6):1762-72.

[221] Liotta F, Gacci M, Frosali F, Querci V, Vittori G, Lapini A, Santarlasci V, Serni S, Cosmi L, Maggi L, et al. Frequency of regulatory T cells in peripheral blood and in tumour-infiltrating lymphocytes correlates with poor prognosis in renal cell carcinoma. BJU Int. 2011;107(9):1500-6.

22. Most references in a review (typically 90% or more) should point to original primary research papers rather than previously published reviews. A review of reviews is not particularly useful to the readers. A few exceptions can be made for outstanding reviews and opinion/conceptual articles. Two examples are listed below but this remark applies for the whole manuscript. I would recommend the authors to replace most reviews in the reference list with appropriate primary research papers.

Please see our answer in the initial comment and comment #2.

23. Page 11, lines 222-223. After “Similar to CD8+ T cells, cytotoxic CD4+ T cells utilise two main cytotoxic pathways; Fas/FasL and Granzyme/Perforin to carry out their cytotoxic mechanisms [7].”, a reference to one or several primary research paper(s) would be much better than to a review.

We have now added a primary reference (Page 11, Line 237)

[82] Kagi D, Vignaux F, Ledermann B, Burki K, Depraetere V, Nagata S, Hengartner H, Golstein P. Fas and perforin pathways as major mechanisms of T cell-mediated cytotoxicity. Science. 1994;265(5171):528-30.

24. Page 13, lines 322-324. After “IL6, IL21, IL23, and TGF-beta are the major signalling cytokines involved in Th17 cells differentiation, and the master regulator is retinoic acid receptor-related orphan receptor g-T (ROR t in humans) [103].”, a reference to a primary research paper would be much better than to a review.

We have now added two primary references (Page 13, Line 350)

[124]   Veldhoen M, Hocking RJ, Atkins CJ, Locksley RM, Stockinger B. TGFbeta in the context of an inflammatory cytokine milieu supports de novo differentiation of IL-17-producing T cells. Immunity. 2006;24(2):179-89.

[125] Yang XO, Panopoulos AD, Nurieva R, Chang SH, Wang D, Watowich SS, Dong C. STAT3 regulates cytokine-mediated generation of inflammatory helper T cells. The Journal of biological chemistry. 2007;282(13):9358-63.

Reviewer 2 Report

This is good comprehensive review on CD4T cell subsets and functions. I would recommend to add more clinical data on use of CD4 T cells in clinical trials.

Author Response

This is good comprehensive review on CD4 T cell subsets and functions. I would recommend to add more clinical data on use of CD4 T cells in clinical trials.

We thank Reviewer #2 for their positive comment on our review. Since this review aims to give a broad and detailed understanding of the six main CD4+ T cell subsets, we have touched upon some clinical applications (Section 6.3). However, we would not be able to give such an important topic the attention or space that is deserves and as such, feel it is outside of the scope of this review. We agree that this would be a very interesting and important review for the field.

Reviewer 3 Report

The review is clear and well written. It can be a good material for students and immunology "beginners" because it describes the CD4+ populations in a simple but clear way and there are appropriate references to facilitate further research in the topic.

Author Response

The review is clear and well written. It can be a good material for students and immunology "beginners" because it describes the CD4+ populations in a simple but clear way and there are appropriate references to facilitate further research in the topic.

We thank Reviewer #3 for their encouraging comment regarding our review. Reviewer #3 did not have criticisms to be addressed.

Round 2

Reviewer 1 Report

The authors have done a great job revising their manuscript. All my concerns were properly addressed.